# Cardiac Radiofrequency Ablation Simulation Using a 3D-Printed Bi-Atrial Thermochromic Model

**Shu Wang** [1,†]**, Carlo Saija** [1,†]**, Justin Choo** [1]**, Zhanchong Ou** [1]**, Maria Birsoan** [1]**, Sarah Germanos** [1]**,
**Joshua Rothwell** [1] **, Behrad Vakili** [1]**, Irum Kotadia** [1]**, Zhouyang Xu** [1]**, Adrian Rolet** [1]**, Adriana Namour** [1]**,
**Woo Suk Yang** [1]**, Steven E. Williams** [1,2,‡] **and Kawal Rhode** [1,*,‡]

[1]  Department of Surgical & Interventional Engineering, School of Biomedical Engineering & Imaging Sciences,
    King's College London, St. Thomas Hospital, London SE1 7EH, UK; shu.1.wang@kcl.ac.uk (S.W.);
    carlo.saija@kcl.ac.uk (C.S.); justin.choo@kcl.ac.uk (J.C.); zhanchong.ou@kcl.ac.uk (Z.O.);
    maria.birsoan@kcl.ac.uk (M.B.); sarah.germanos@kcl.ac.uk (S.G.); joshua.rothwell@kcl.ac.uk (J.R.);
    behrad.vakili@kcl.ac.uk (B.V.); irum.kotadia@kcl.ac.uk (I.K.); zhouyang.xu@kcl.ac.uk (Z.X.);
    adrian.rolet@kcl.ac.uk (A.R.); adriana.namour@kcl.ac.uk (A.N.); woo.s.yang@kcl.ac.uk (W.S.Y.);
    steven.e.williams@kcl.ac.uk (S.E.W.)
[2]  Centre for Cardiovascular Science, The University of Edinburgh, Edinburgh EH8 9YL, UK
[*]  Correspondence: kawal.rhode@kcl.ac.uk
[†]  These authors contributed equally to this work.
[‡]  These authors contributed equally to this work.

**Featured Application: The simulator has applications in the training of electrophysiologists for cardiac radio-frequency ablation therapy and the evaluation of novel cardiac ablation devices.**

**Abstract:** Radiofrequency ablation (RFA) is a treatment used in the management of various arrhythmias including atrial fibrillation. Enhanced training for electrophysiologists through the use of physical simulators has a significant role in improving patient outcomes. The requirements for a high-fidelity simulator for cardiac RFA are challenging and not fully met by any research or commercial simulator at present. In this study, we have produced and evaluated a 3D-printed, bi-atrial model contained in a custom-made enclosure for RFA simulation using a new soft tissue-mimicking polymer, Layfomm-40, combined with thermochromic pigment and barium sulphate in an acrylic paint carrier. We evaluated the conductive properties of Layfomm-40, its sensitivity to RFA, and its visibility in X-ray imaging, and carried a full simulation of RFA in the cardiac catheterization laboratory by an electrophysiologist. We demonstrated that a patient-specific 3D-printed Layfomm-40 bi-atrial model coated with a custom thermochromic/barium sulphate paint was compatible with the CARTO3 electroanatomic mapping system and could be effectively imaged using X-ray fluoroscopy. We demonstrated the effective delivery and visualization of radiofrequency ablation lesions in this model. The simulator meets nearly all the requirements for high-fidelity physical simulation of RFA. The use of such simulators is likely to have impact on the training of electrophysiologists and the evaluation of novel RFA devices.

**Keywords:** electrophysiology; cardiac radiofrequency ablation; 3D-printing; Layfomm-40; physical simulation; simulation training; thermochromic pigments

## 1. Introduction

Cardiac ablation therapy is a minimally invasive interventional procedure used in the treatment of cardiac arrhythmias. The process involves the insertion of flexible catheters through peripheral blood vessels, which are guided to the site of abnormal electrical conduction in the myocardium. Here, the arrhythmia is terminated via destruction of the pathological tissue, most commonly using radiofrequency ablation (RFA). The procedure is performed in the cardiac catheterization laboratory under X-ray fluoroscopy guidance and

often with the use of an electroanatomic mapping system (EAMS), which tracks the inserted catheters and allows measurement of the patient's cardiac anatomy and electrophysiology. Hong et al. describe the current strategies and technologies for the ablation of the most common arrhythmia, atrial fibrillation [1].

RFA procedures are complex and require considerable training, sometimes delivered via computer or physical simulators. Computers and physical simulators have an ethical and cost advantage but may lack fidelity. An example of a computer simulator is the Mentice VIST (Mentice AB, Göteborg, Sweden), which has been shown to improve electrophysiology-trainee performance [2]. The requirements for a high-fidelity physical simulator for cardiac RFA make it challenging to produce such a simulator. These requirements include representation of the cardiac anatomy, soft material properties, realistic appearance under imaging (particularly X-ray fluoroscopy), compatibility with EAMSs, sensitivity to RFA, and generation of electrophysiological signals. We are not aware of any physical simulator that meets all these criteria. However, several simulators have been developed that meet a subset of the criteria. Rossi et al. [3] developed a physical simulator using a 3D-printed whole heart model embedded into a custom torso. The heart was printed using thermoplastic polyurethane and the simulator was compatible with the CARTO3 EAMS (Biosense Webster, Irvine, CA, USA). The simulator was evaluated by 10 electrophysiologists and used to compare novel to experienced operators. Similar simulators are produced by Heartroid (JMC Corporation, Yokohama, Japan, https://www.heartroid.com/itemlist/ablation/, accessed on 1 December 2021) and Pangolin (Tel-Aviv, Israel, http://pangolin.co.il/en/gallery/simulators-endo-vascular/, accessed on 1 December 2021). However, none of these physical simulators have sensitivity to RFA, meaning that ablation lesions cannot be created in these simulators and as a result, effects of intended therapies delivered by trainees cannot be quantified. Lesion formation depends on the parameters of the RFA, such as power, duration, ablation temperature and importantly, the contact force between the ablation catheter and the tissue [4]. Therefore, a simulator that allows creation of lesions will add a valuable layer to assessing a trainee's progression.

Several attempts have been made to create a physical simulation medium that is sensitive to RFA and therefore able to demonstrate lesions. Bu-Lin et al. [5] used a polyacrylamide gel with bovine albumin which produced a noticeable color change after RFA due to coagulation between 50 and 60 °C. Negussie et al. [6] proposed the use of thermochromic pigments to create RFA-sensitive models that produced a permanent color change above 60 °C. However, there have been no attempts to incorporate ablation-sensitivity into a cardiac RFA simulator.

3D-printed models are widely used in the medical field for a variety of purposes. Current applications include, but are not limited to, implant and prosthetic design, biomedical device testing, and, notably, pre-operative/procedural planning and surgical/interventional simulation training [7]. Furthermore, advances in the field of additive manufacturing have facilitated the production of objects with complex geometries. Notably, the incorporation of patient scans as the basis for model design enables the fabrication of patient-specific simulation aids bearing greater anatomical accuracy. This is particularly relevant in the rehearsal of complex surgical/interventional procedures, also allowing for anatomical variation between patients. In our previous work, we evaluated the use of 3D-printed thermoplastics for creating patient-specific whole heart models that were multimodal-imaging-compatible [8]. We investigated a low-cost, soft-tissue-mimicking copolymer filament, known as Layfomm-40 from the Poro-Lay series (CC-Products, Köln, Germany) [9,10]. Layfomm-40 filament is rigid and consists of polyvinyl alcohol (PVA) and a thermoplastic elastomer (TPE) composite. This allows the material to be 3D printed using a fusion deposition modelling (FDM) printer at low cost. Once the 3D print is soaked in water, the PVA dissolves and leaves a spongey, microporous TPE composite which mimics soft tissue. In our work, we found Layfomm-40 to be an excellent material for creating cardiac models in terms of soft material properties and imaging properties [8].

The aim of this work was to evaluate the use of Layfomm-40 for creating a physical cardiac RFA simulator that could satisfy as many as possible of the requirements mentioned previously. We investigated the electrical and thermal conductivities of Layfomm-40 to ensure compatibility with RFA and EAMSs, investigated the use of thermochromic pigments for RFA-sensitivity, and investigated techniques for optimum visibility of Layfomm-40 under X-ray fluoroscopy. We integrated our findings to produce a bi-atrial, patient-specific model that was housed in a custom enclosure, and we tested this simulator in the cardiac catheterization laboratory environment to prove the overall concept.

## 2. Materials and Methods

### 2.1. Layfomm-40 Conductivity Analysis

For a 3D-printed Layfomm-40 model to be compatible with RFA and EAMSs, it must be both electrically and thermally conductive. We 3D printed five 10 mm cubes of Layfomm-40 and immersed these in saturated saline solution for 3 days. The concentration of the saline solution affects conductivity and using a saturated solution gave the maximum conductivity achievable. Electrical conductivity testing was completed using an AstroAI multimeter (AstroAI, Garden Grove, CA, USA) using the method shown in Figure 1a. The conductivity, $\sigma$, was calculated using Equation (1), where $R$ denotes electrical resistance, $A$ denotes the cross-sectional area, and $L$ denotes the current path length.

$$\sigma = \frac{L}{R \times A} S \cdot m^{-1} \tag{1}$$

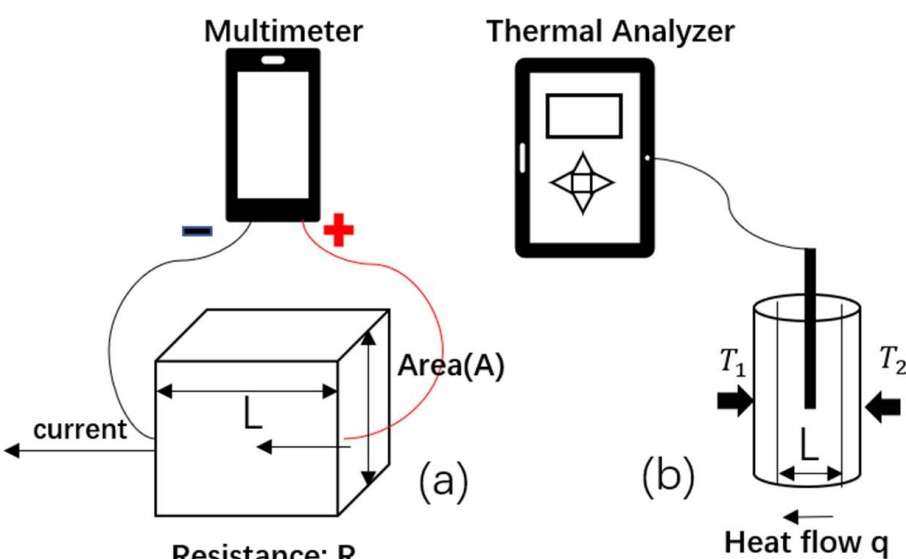

**Figure 1.** Illustration of (**a**) electrical conductivity measurement using a multimeter and (**b**) thermal conductivity measurement using a thermal property analyzer.

The thermal conductivity measurement was conducted with a KD2 Pro Thermal Analyzer (Decagon, Pullman, WA, USA) using the method shown in Figure 1b. The prepared samples of Layfomm-40 were 100 mm long, 15 mm thick, and had a 2.4 mm diameter lumen to insert the needle probe. Five samples were immersed in saturated saline solution for 3 days. The thermal conductivity was calculated using Equation (2), where q denotes the heat flow, $\kappa$ denotes the thermal conductivity, $T_2 - T_1$ corresponds to the temperature difference, and L represents the travel distance of heat flow.

$$q = \kappa \cdot \frac{T_2 - T_1}{L} \tag{2}$$

## 2.2. Thermochromic Paint Formulation and RFA-Sensitivity

Thermochromic pigments display a color-changing effect, which is induced by heating or cooling [11,12]. In terms of whether the original color can be recovered, thermochromic materials can be classified as reversible or irreversible. Reversible pigments are usually composed of a color developer, a color former, and a co-solvent. On heating, the co-solvent changes from solid to liquid and allows the former to separate from the developer, leading to the color change. On cooling, the process is reversed (Figure 2a). Irreversible pigments have a volatile dye that evaporates when heated to the phase transition temperature, resulting in a permanent colorless state (Figure 2b).

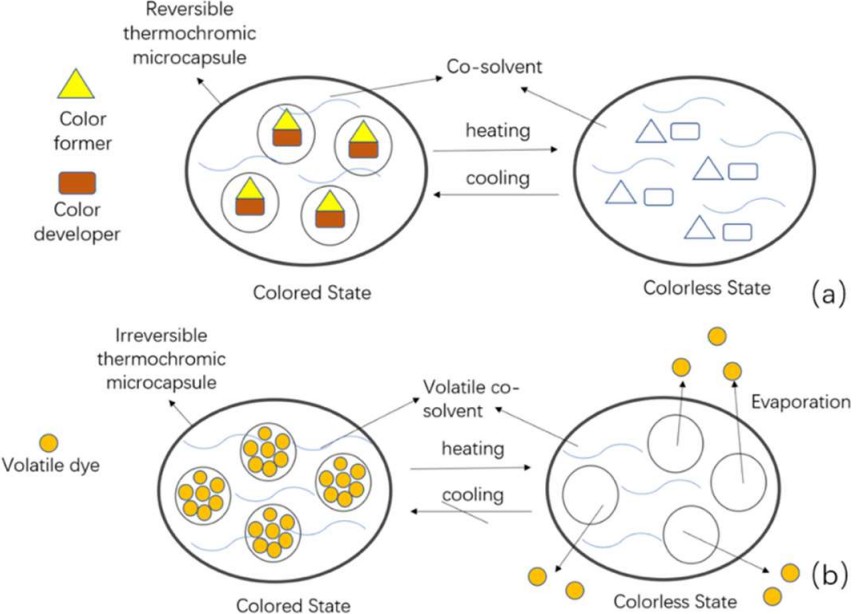

**Figure 2.** Mechanisms of action of thermochromic microcapsules during heating and cooling. (**a**) Reversible and (**b**) irreversible.

Since we want the color change to be permanent and occur at typical temperatures required for RFA lesion formation, we selected an irreversible thermochromic pigment with a transition temperature of 60 °C. This pigment was black below the transition temperature and colorless above this temperature (Special FX Creative, Newhaven, UK, https://www.sfxc.co.uk/products/sfxc-irreversible-thermochromic-pigment-60-c-black, accessed on 10 November 2021). The pigment was mixed with (white) unpigmented acrylic paint (UAP) (10:1 UAP:pigment mass ratio) to form a grey thermochromic paint that could be applied to the 3D-printed Layfomm-40.

Several discs (radius 20 mm and thickness 5 mm) of Layfomm-40 were printed and painted with the thermochromic paint. These were soaked in 0.9 w/w saline solution and then tested for RFA-sensitivity. The discs were immersed in a saline bath (~38 °C) and radiofrequency was delivered through a Stockert 70 Cardiac Ablation RF Generator (Biosense Webster, USA) and a non-irrigated ablation catheter. The catheter tip was kept orthogonal to the disc plane. Ablations of 60 s duration were applied while varying the power (30–90 W) and temperature (50–80 °C) settings. Only one ablation was applied for each combination of power and temperature. Ablations were also applied to a piece of chicken breast for comparison. The diameter of the lesions was measured for each setting with a digital caliper. Once parameters for consistent size lesion creation were identified, the experiment was repeated using a $50 \times 50 \times 2$ mm$^3$ sample of thermochromic-paint-coated Layfomm-40 which was divided into nine subsections. Nine ablations were applied keeping the power and temperature settings between 70–80 W and 70–80 °C, respectively. The lesion sizes were measured, and the average size computed for these power/temperature settings.

### 2.3. Increasing X-ray Visibility

In our previous work [8], we found the X-ray attenuation of Layfomm-40 cardiac models was only marginally different to that of water. Therefore, since the models need to be immersed in a water tank as part of the simulation, these un-modified models are not easily visualized in X-ray fluoroscopy. We investigated the use of barium sulphate ($BaSO_4$) to increase the model visibility. $BaSO_4$ is a commonly used X-ray contrast agent and is often incorporated into polymers that need to be X-ray visible. We doped UAP with $BaSO_4$ in a range of mass ratios from 5:2 (UAP:$BaSO_4$) to 15:1. Hollow tubes of Layfomm-40 were prepared and coated with the different doped UAP formulations. These were soaked in saline for 24 h. These were then immersed in a tank filled with 0.9% $w/w$ saline and imaged using our cardiac catheterization laboratory (Siemens Artis Q Biplane, Siemens Healthineers, Erlangen, Germany) using our standard electrophysiology X-ray protocol. We exported the captured image data and analyzed the maximum percentage image contrast between the walls of the sample (Layfomm-40 with doped UAP) and the background (saline) based on region-of-interest analysis. This analysis was performed with ImageJ (https://imagej.nih.gov/ij/index.html, accessed on 17 November 2021). For reference, we calculated the percentage contrast between the cardiac shadow and the surrounding lung tissue in representative clinical X-ray fluoroscopy images taken from patients in the same catheterization laboratory using the same protocol.

### 2.4. Atrial Model and Simulator Build-Up

Having verified the RFA-sensitivity and formulated the thermochromic and $BaSO_4$-doped paints, we proceeded to develop the complete simulator. To ensure reproducibility and cost-effectiveness of the simulator, most parts were manufactured using FDM or stereolithography (SLA) 3D-printing. The simulator comprises four main components: the cardiac model, its base, a transparent tank and its lid (with patch holders).

The computer model of the atria and associated great vessels was designed via medical image segmentation and processing. An adult male contrast-enhanced chest computer tomography (CT) scan was segmented using the semi-automatic segmentation feature of ITK-SNAP (University of Pennsylvania, Philadelphia, PA, USA). Following this initial step, the segmentation was refined using the smoothing tool (Level 2) in Seg3D (University of Utah, Salt Lake City, UT, USA). Subsequently, the dilation-erosion function was applied to create an inner mold of this segmentation. The final hollow model was the differential result between the segmentation and the inner mold, giving a wall thickness of 1 mm. This model was exported to Fusion360 (Autodesk, San Rafael, CA, USA) and extruded by 1.5 mm to give an overall wall thickness of 2.5 mm. The pulmonary veins were cut to a length of 20 mm. The venae cavae were cut and lofted to a standard-diameter tube fitting size (inner diameter: 16 mm, outer diameter: 23 mm). The model was divided into two sections for 3D printing to allow for the use and easy removal of support material. The exported meshes were sliced using Cura (Ultimaker, Utrecht, The Netherlands) and manufactured using a Chiron FDM 3D-printer (Anycubic, Shenzhen, China). A transeptal puncture was made in the atrial septum for right-to-left access. Paint was applied to the inner surface of model in four layers with a 3 h drying time between each layer. Following this, the two sections of the model were welded together using a digital soldering iron, ensuring not to transfer heat to the inner coats of paint. Finally, the outside of the model was coated with paint in a similar manner to the inside.

The simulator system was contained within a $48 \times 39 \times 31$ cm transparent box (Really Useful Plastic Box, Badford, UK) with catheter entry points connecting to each of the venae cavae via silicone tubing. The entry point for the inferior vena cava was extended outside the box using silicone tubing to simulate realistic pathlength from femoral venous access sites. The box was filled to a depth of 25 cm with 38 °C 0.9% w/w saline solution to simulate the human thorax and its conductive properties. The lid of the box was designed to allow compatibility with the CARTO3 EAMS. Six tubes were integrated into the lid to accommodate the six CARTO3 patches, with three patches above the heart

model (simulating the patches on a patient's anterior chest wall) and three at the bottom (simulating the patches on the posterior chest wall). The tubes were constructed using standard polyvinylchloride (PVC) pipes and gasketed screw-on end-cap fittings with polytetrafluoroethene tape to ensure no leaks were present. The model base was designed using Fusion360 to hold the model in place during the simulation and 3D printed using a Photon SLA printer (Anycubic, China). The model rests on the base with a foam insert, custom clamps hold the venae cavae, and a hook-and-loop strap goes around the body of the model. This arrangement allows models with variable geometry to be firmly held in the simulator and to be inserted and removed easily.

### 2.5. X-ray Imaging, Mapping, and Ablation

The simulator was taken to the cardiac catheterization laboratory, placed on the patient table and connected to the CARTO3 system. X-ray fluoroscopy imaging was performed at several standard view angles using the standard electrophysiology protocol on the system. A cone beam CT scan was performed using the standard protocol.

An 8F ThermoCool SmartTouch SF (Biosense Webster, USA) ablation catheter was inserted via an 11 cm 8F introducer sheath (Cordis, Santa Clara, CA, USA) into the inferior vena cava. The ablation catheter was connected to the CARTO3 system, the irrigation system, and the RF generator. The right side of the model was mapped to generate the geometry of the right atrium and the venae cavae. The introducer sheath was replaced with a 60 cm 8.5F 55° Heartspan Transseptal Fixed Sheath (Biotronik, Berlin, Germany). This was manipulated under X-ray fluoroscopy guidance to pass the ablation catheter into the left atrium through the transseptal puncture in the heart model. Mapping was then performed of the left atrium.

Ablations were performed only in the right side of the heart model. A line of five ablations was performed at the superior aspect of the posterior intercaval line with increasing ablative power. From inferior to superior, the power settings for each ablation point were as follows: 15.9 W; 16.0 W; 22.0 W; 23.0 W; and 30.6 W. The ablation duration was fixed to 45 s. Subsequently, using a fixed power setting of 21.5 W and the same duration, a line of five ablations was performed in the posterior wall of the right atrium. The heart model was then removed from the simulator and cut open to examine and measure the lesions.

## 3. Results

### 3.1. Layfomm-40 Conductivity Analysis

The electrical conductivity of Layfomm-40 immersed in saturated saline solution was determined to be from $1.3 \times 10^{-7}$ to $3.0 \times 10^{-6}$ S/m. The thermal conductivity was determined to be from 0.34 to 0.45 W/m/K. In comparison, literature values for myocardium are 0.16 S/m [13] and 0.56 W/m/K [14], respectively. The electrical conductivity was much lower than the physiological value, but the thermal conductivity was similar.

### 3.2. Thermochromic Paint Formulation and RFA-Sensitivity

The Layfomm-40 discs with thermochromic paint coating were confirmed to be RFA-sensitive. Table 1 shows that lesions were formed on the discs with temperatures ≥60 °C and power settings ≥40 W. The diameter of the lesions best matched those in the chicken breast (Figure 3a) when the temperature and power were ≥70 °C and ≥70 W, respectively. Settings below these either produced no lesions or inconsistently sized lesions. Using temperatures and powers above 90 °C and 90 W could produce a maximal lesion diameter of approximately 7 mm (Figure 3b). Consistent lesions were produced with temperatures of 70–80 °C and powers of 70–80 W (Figure 3c). The average lesion diameter for these settings was measured to be 3.3 ± 0.3 mm (±1 SD, n = 9).

**Table 1.** Ablation lesion sizes (mm) on thermochromic Layfomm-40 discs using different ablation temperatures and power settings. The ablation duration was 60 s.

| Ablation Temperature (°C)/Power (Watts) | 50 °C | 60 | 70 | 80 |
|---|---|---|---|---|
| 30 W | NA | NA | NA | NA |
| 40 | NA | NA | 2.7 | 2.4 |
| 50 | NA | 2.3 | 1.3 | 1.6 |
| 60 | NA | 1.4 | 1.8 | 3.3 |
| 70 | NA | 2.4 | 3.2 | 3.3 |
| 80 | NA | 1.9 | 3.0 | 3.2 |
| 90 | NA | 2.5 | 3.2 | 3.5 |

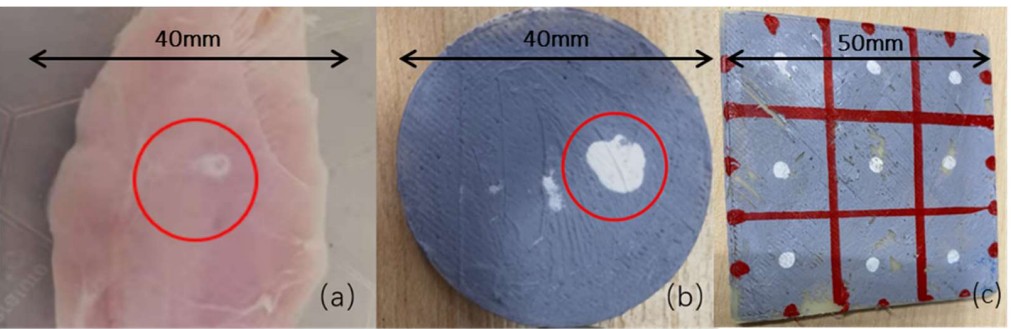

**Figure 3.** (**a**) Ablation lesion in chicken breast at 70 W, 70 °C, 60 s with a lesion diameter of ~3 mm, (**b**) Layfomm-40 disc with thermochromic paint coating showing the maximum size lesion that could be created at >90 W, >80 °C, 60 s with a lesion diameter of ~7 mm, and (**c**) Layfomm-40 square with thermochromic paint coating showing nine lesions at 70–80 W, 70–80 °C, 60 s with lesion diameters of ~3 mm.

*3.3. Increasing X-ray Visibility*

Figure 4 shows the effect of increasing the concentration of $BaSO_4$ in the acrylic paint on X-ray image contrast. Table 2 show the calculated maximum percentage image contrast for the different mass ratios with a value computed from clinical image data for comparison. It was found that the 5:1 mass ratio gave a contrast that best matched what was seen in clinical images.

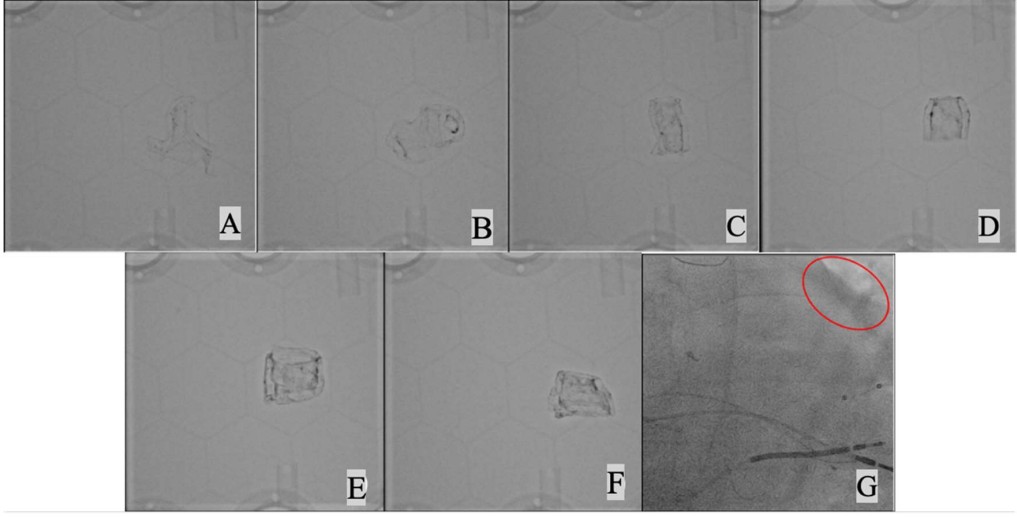

**Figure 4.** X-ray imaging of Layfomm-40 cylinders coated with UAP mixed with increasing amounts of $BaSO_4$. (**A**) mass ratio15:1 UAP:$BaSO_4$, (**B**) 15:2, (**C**) 5:1, (**D**) 15:2, (**E**) 3:1, (**F**) 5:2, and (**G**) clinical image showing the left heart border (red oval).

**Table 2.** The maximum percentage image contrast for $BaSO_4$–UAP coated Layfomm-40 cylinders at different UAP:$BaSO_4$ mass ratios compared to myocardial contrast achieved in clinical images.

| UAP:$BaSO_4$ | 15:1 | 15:2 | 5:1 | 15:4 | 3:1 | 5:2 | Myocardium |
|---|---|---|---|---|---|---|---|
| **% Contrast** | 5.56 | 15.6 | 23.6 | 28.2 | 35.5 | 40.0 | 21.0 |

Combining the findings from 3.2 and 3.3, a strategy was formulated for the coating of Layfomm-40 for the heart model. Figure 5 illustrates the architecture of the coating. There are four layers of paint applied both externally and internally consisting of a sandwich of UAP, UAP doped with $BaSO_4$ and UAP doped with thermochromic pigment.

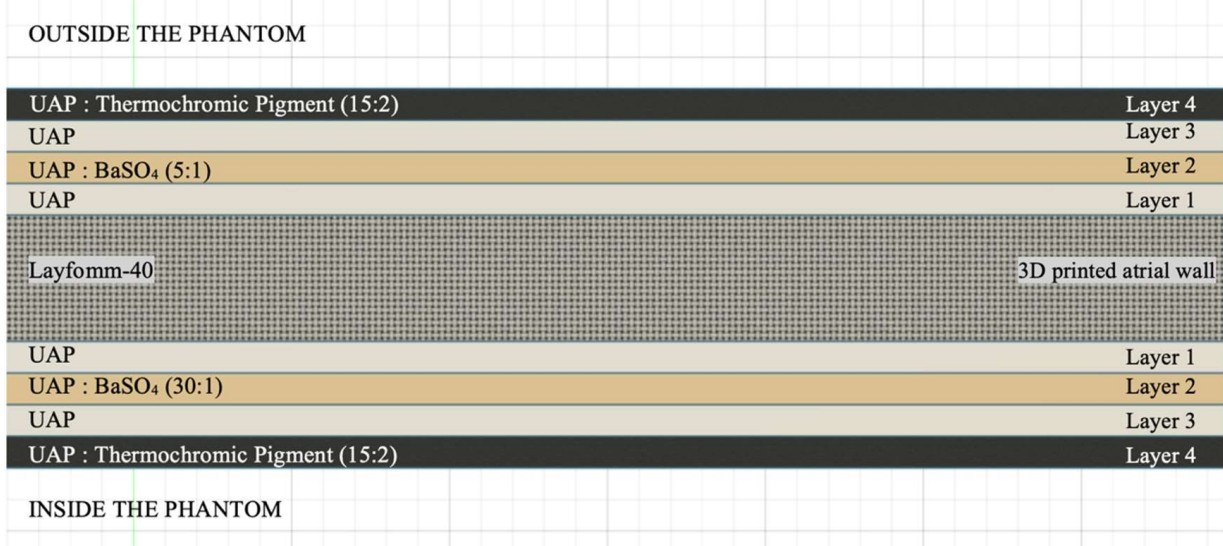

**Figure 5.** Architecture of custom paint coatings of the Layfomm-40 model.

### 3.4. Atrial Model and Simulator Build-Up

Figure 6 shows the solid and hollow computer model that was generated by image segmentation and processing of the patient CT data. Figure 7 shows the steps to produce the bi-atrial model from the computer model.

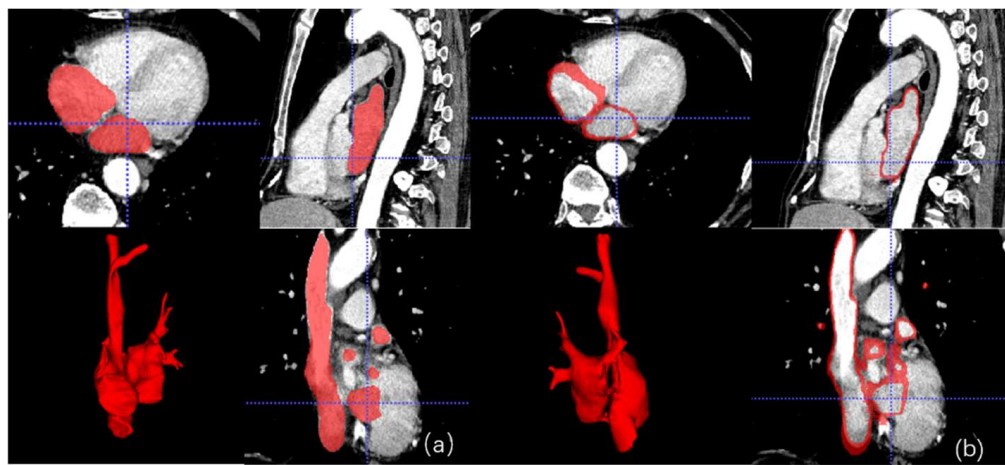

**Figure 6.** (**a**) Solid and (**b**) hollow bi-atrial computer model showing the segmentation in multiplanar views and a 3D rendering.

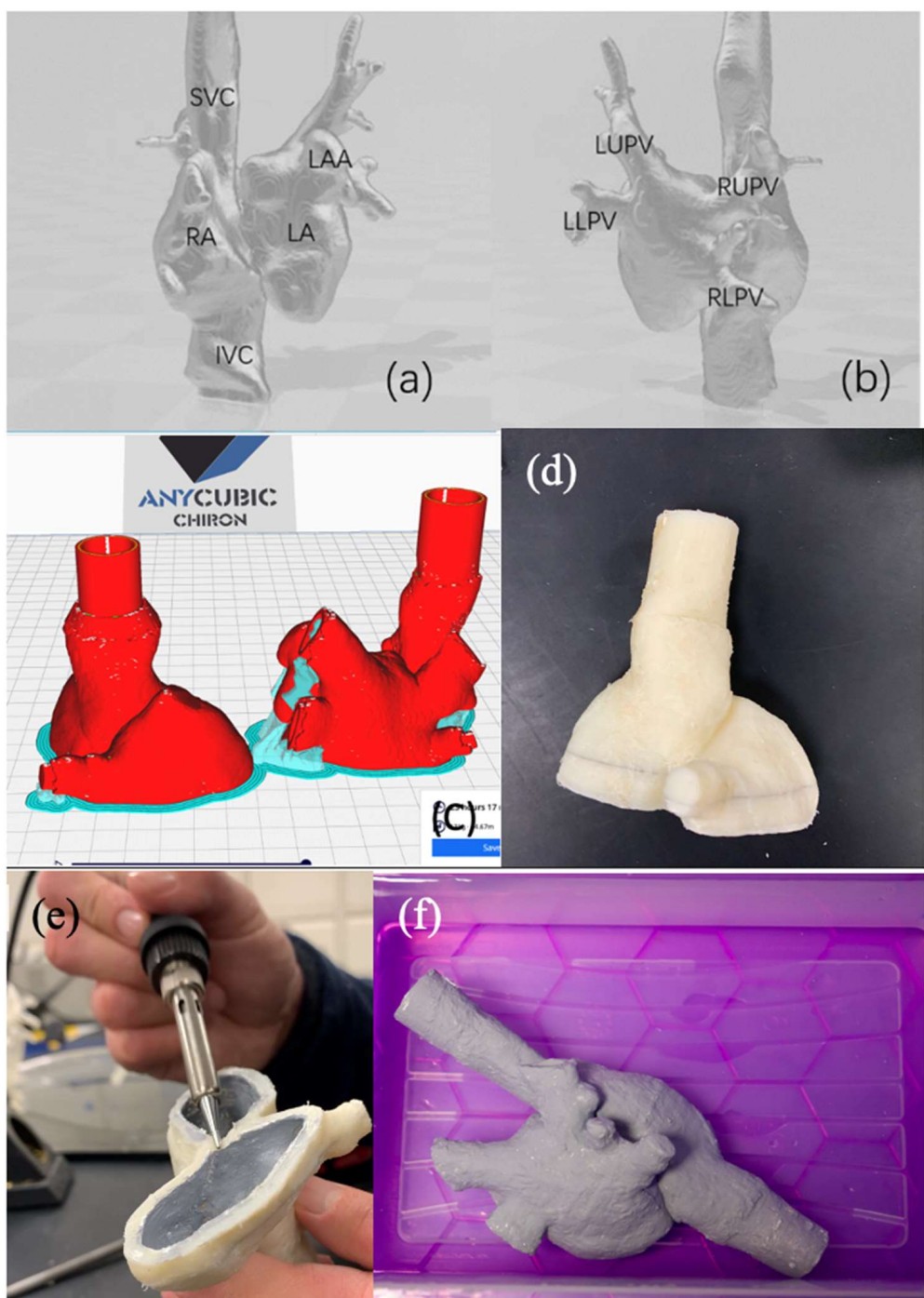

**Figure 7.** Bi-atrial model rendering from (**a**) anterior view and (**b**) posterior view. (**c**) Bi-atrial model cut into two sections for ease of printing and coating. (**d**) Lower section printed in Layfomm-40. (**e**) Lower section coated internally with paint layers and ready for plastic welding. (**f**) Fully welded and external-paint-coated bi-atrial model soaked in saline solution. SVC—superior vena cava, IVC—inferior vena cava, RA—right atrium, LA—left atrium, LAA—left atrial appendage, LUPV—left upper pulmonary vein, LLPV—left lower pulmonary vein, RUPV—right upper pulmonary vein, and RLPV—right lower pulmonary vein.

Figure 8 shows the features of the simulator base and enclosure. Arrangement of the CARTO3 patches can be seen as well as the methods used to allow entry of devices into the heart model.

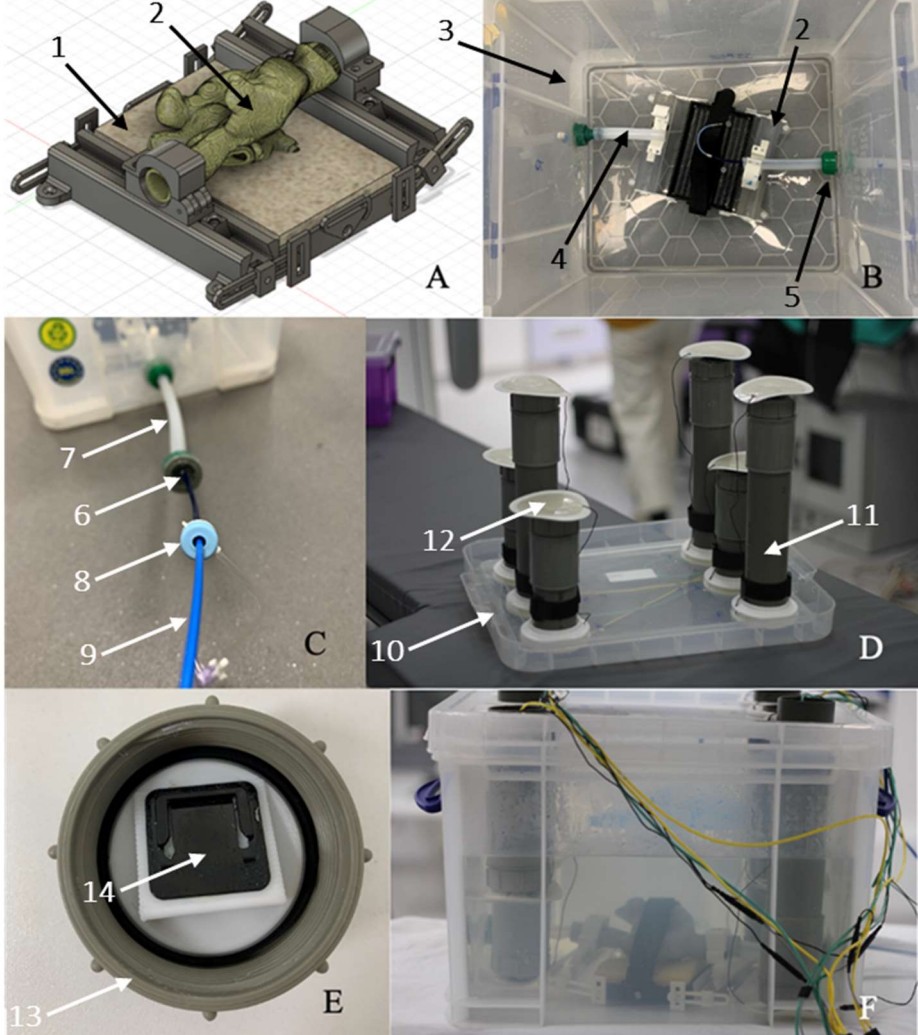

**Figure 8.** (**A**) Fusion360 model of simulator base (1) with inserted bi-atrial model (2). (**B**) Simulator base inside the Really Useful Box (3) with connecting silicone tubing (4) leading to standard hose connectors (5) with silicone plugs (6) to act as hemostatic valves. (**C**) Silicone tubing extension (7) to IVC entry point showing insertion of a sheath (8) and ablation catheter (9). (**D**) lid assembly (10) with PVC tubes (11) to accommodate the CARTO3 patches (12). (**E**) PVC tube end cap (13) with CARTO3 patch connector (14). (**F**) Complete simulator assembly with wiring to CARTO3 system and RF generator.

### 3.5. X-ray Imaging, Mapping, and Ablation

Figure 9 show examples of X-ray fluoroscopy images taken during the simulated procedure. The bi-atrial model is clearly visible and there is good epicardial contrast between the Layfomm-40 and the surrounding saline solution. The cone beam CT image (512 × 512 × 400 matrix size, 0.46 mm$^3$ voxel size) was used to verify the dimensions of the heart model. The overall model size was 72 × 66 × 160 mm. The LA dimensions were 38–44 mm, which fall within the typical range [8]. The RA dimensions were 29–35 mm, which also fall within the typical range [15]. Figure 10 shows the exported chamber geometries and ablation points from CARTO3 and the corresponding lesions formed on the heart model. During the first five ablations, the ablative power was increased gradually from inferior to superior and there was a clear corresponding effect on the lesion size. For the final five ablations, the ablative power was kept constant, and the mean lesion diameter was measured to be 3.4 ± 0.6 mm (±1 SD, n = 5). The results showed that the RFA only

affected the internal coatings of the heart model and had no effect transmurally or on the outer coatings.

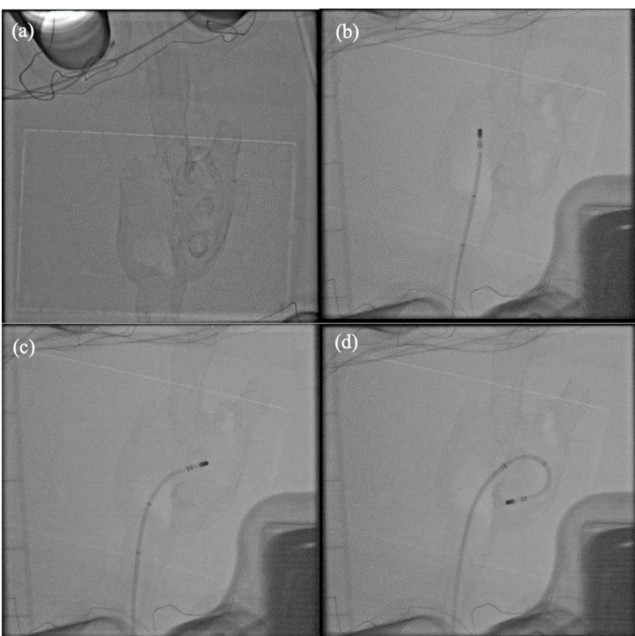

**Figure 9.** X-ray fluoroscopy imaging of the simulator. (**a**) With no catheters inserted, (**b**) ablation catheter in the RA, (**c**) ablation catheter across the septum, and (**d**) ablation catheter with transseptal sheath in the LA.

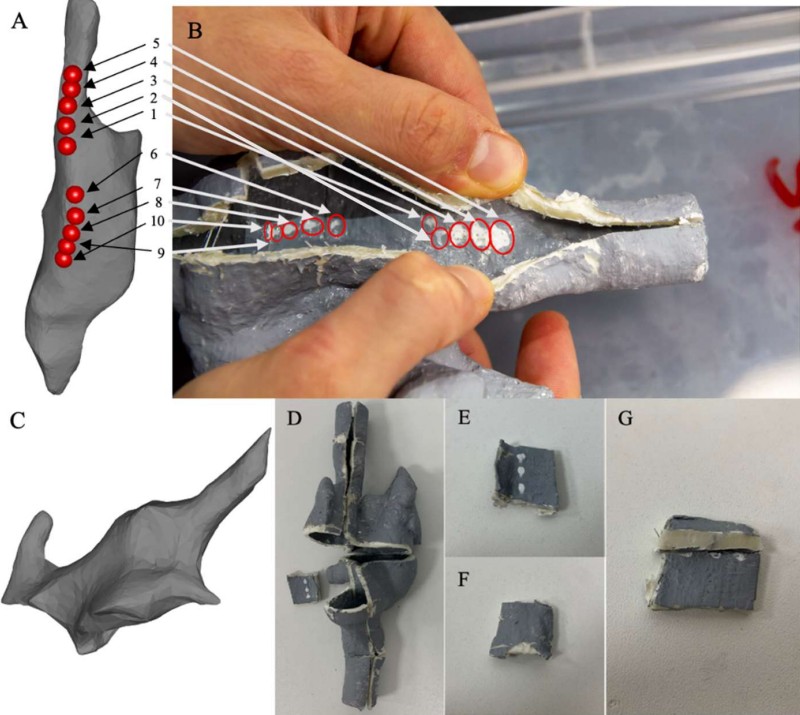

**Figure 10.** CARTO3 mapping and ablation. (**A**) Exported geometry and ablation points (numbered 1 to 10) from the CARTO3 system showing the RA and the venae cavae. (**B**) Cut-away of the heart model immediately after the ablation experiment showing the lesions and their correspondence to the ablation points. (**C**) Partial CARTO3 map of the LA. (**D**) Cut-away of the dry heart model. (**E**) Ablation points clearly visible on the internal surface. (**F**) External surface is not affected by the RFA. (**G**) Transmural cut that shows no effect inside the model wall.

## 4. Discussion

The aim of this study was to produce a novel 3D-printed, irreversible thermochromic bi-atrial model, meeting as many of the requirements for a high-fidelity physical simulator for cardiac RFA as possible. We combined Layfomm-40, a new tissue-mimicking 3D-printable polymer, with an irreversible thermochromic pigment and barium sulphate using acrylic paint as a carrier. We measured the electrical and thermal conductivities of Layfomm-40 soaked in saline solution. The thermal properties were found to be similar to those of the myocardium. Although the electrical conductivity was substantially lower it was sufficient to allow use of the material with RFA and EAM systems. We developed an irreversible thermochromic paint and demonstrated that Layfomm-40 coated with this paint was sensitive to RFA and produced visible lesions whose size was influenced by the power and temperature settings of the ablation system. Lesions with diameters similar to those produced in isolated muscle and in vivo could be generated. We overcame issues with the X-ray visibility of Layfomm-40 by formulating a barium-sulphate-doped acrylic paint that produced a similar tissue-to-background image contrast to that seen in clinical cardiac X-ray fluoroscopy images. The thermochromic paint and the barium-doped paint were combined into a four-layer coating for applying to Layfomm-40, both internally and externally. A bi-atrial model was constructed using these strategies, starting with a patient CT scan. A custom base and enclosure were designed and constructed that allowed insertion of catheters, stabilized the model during catheter manipulation and allowed compatibility with standard RF generators and the CARTO3 EAMS. Experimental validation of the simulator by a cardiologist demonstrated the successful ability to perform X-ray fluoroscopy and cone beam CT imaging, to insert and exchange interventional devices without leaks, to manipulate devices within the bi-atrial model without causing damage, to perform mapping using CARTO3, and to perform ablations. Inspection of the cut away model after the experiment clearly showed the ablation lesions.

The striking novel feature of our simulator is the ability to visualize delivered radiofrequency ablation lesions. To the best of our knowledge a common limitation of all prior physical simulators is the inability to visualize the delivered therapy. The state-of-the-art simulators developed by Rossi et al. [3], Heartroid, and Pangolin are limited by not being RFA-sensitive. Since the goal of electroanatomic mapping systems is to guide the delivery of radiofrequency ablation therapy we argue that visualization of this therapy is an essential requirement to support effective simulator-based training of electrophysiologists. In this work we have developed a simple but effective methodology supporting this visualization which can be easily applied to any 3D-printed cardiac chamber model which potentially could be of value during training or device evaluation scenarios. As mentioned previously, there are many factors that affect RFA lesion formation [4]. For lesions 1–5 shown in Figure 10B, we see that increasing power increased the lesion diameter while keeping all other parameters constant, which was as expected. One parameter which is difficult to keep constant is the contact force and this ranged from an average (over the time of each ablation) of 11.1 g to 21.7 g during these five ablations. Masnok and Watanabe conducted experiments to investigate the effect of varying contact force on RFA lesion formation in an in vitro set up [16]. They measured the surface lesion diameter, the intramural width, and the intramural depth. Although they used different power and temperature settings (30 W, 30 °C) for their irrigated ablation catheter compared to our experiments, the average surface lesion diameters that they measured were 4.1 mm (for 2 g force) to 6.9 mm (40 g), with our measurement being 3.4 mm (19.7 g average, 14.4–23.4 g range for lesions 6–10 (Figure 10B)). One limitation of our current model is that we cannot see lesion formation intramurally. In fact, the inner layer of doped paint was unaffected by the RFA. It is desirable to create a lesion that extends transmurally but one that does not lead to risk of perforation of the myocardium [16]. Since the doped paint does not penetrate the wall of our model, we cannot see the intramural effects. This is a limitation that could be addressed in future work.

Physical simulators, such as the one presented, are a way of implementing the 3Rs principle—Replacement, Reduction, and Refinement. By performing more humane animal experiments [17], this simulator falls into the Replacement category. Furthermore, although we have not performed a detailed cost analysis, we estimate that the cost of parts to construct our simulator is less than USD 1000 (including suitable SLA and FDM printers and not including labor costs). This compares favorably with the simulator of Rossi et al. which was estimated to have a parts cost of circa USD 7500 [3]. The cost of using animal models or cadavers would also be several thousands of dollars per unit, not taking into account the cost of the specialist facilities that are required to support this type of work. Therefore, our proposed solution is not only cost-effective but also has an ethical advantage.

## 5. Conclusions

Our novel simulator meets many of the requirements for a fully functional physical simulator for cardiac RFA. We believe that it is currently the most comprehensive example of such a simulator. Anatomically accurate, 3D-printed tissue-mimicking thermochromic models, as presented in this paper, may prove to be a reliable, inexpensive, and clinically useful tool in simulating cardiac catheter ablation for either training of healthcare professionals or evaluation of novel RF ablation devices. These provide a valuable alternative to computer simulations, animal models, or cadavers.

## 6. Future Work

In this work we focused on the technical aspects of the simulator and reached a proof-of-concept stage. Future work will focus on evaluating the simulator using a cohort of electrophysiologists and performing standard ablation strategies such as pulmonary vein isolation or wide area circumferential ablation for atrial fibrillation. Several simulators mentioned in the introduction have flow capability. We have not tested this feature in our simulator but there is no reason to believe that the simulator would not be flow-compatible and we aim to develop this feature. Electrophysiologists rely on electrophysiology signals to guide their treatment strategies and currently our simulator is not capable of simulating this. This feature could also be incorporated either via computer-based simulation or physically and this is another area for future investigation.

**Author Contributions:** All authors contributed to conceptualization of methods, carrying out experiments, analysis of results, and writing of the manuscript. A.R. was responsible for the photography. All authors have read and agreed to the published version of the manuscript.

**Funding:** This work was supported by the National Institute for Health Research Biomedical Research Centre at Guy's and St. Thomas' NHS Foundation Trust and King's College London, the Wellcome/EPSRC Centre for Medical Engineering (WT 203148/Z/16/Z) and the KCL-China Scholarship Scheme. The views expressed are those of the authors and not necessarily those of the NHS, the NIHR, or the Department of Health. SEW is supported by the British Heat Foundation (FS/20/26/34952).

**Informed Consent Statement:** Not applicable.

**Data Availability Statement:** Not applicable.

**Conflicts of Interest:** The authors declare no conflict of interest.

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
