# Peer review of "Cardiac Radiofrequency Ablation Simulation Using a 3D-Printed Bi-Atrial Thermochromic Model"

_applsci, doi:10.3390/app12136553_

Round 1
Reviewer 1 Report
The article describes a novel way to evaluate Cardiac Radiofrequency Ablation Simulation by using a 3D-Printed Thermochromic Model. The article is quite interesting, and it can be a big contribution to the Scientifics involved in this research topic. However, in order to improve it, some issues must be addressed.
- Title: I suggest a title that better suits the paper, especially the section proof of concept that can be eliminated.
- Two different symbols to describe electrical resistivity are used (ρ).
- Conductivity analysis: the authors said that the Layfomm-40 were immersed (or emmersed?) in a saturated saline solution for 3 days. As far as I understand, this was done in order to modify the electrical and thermal conductivity of each sample. However, it wasn’t explaining why a saturated saline solution was used instead of playing with different concentrations of NaCl in order to reach the values reported in literature. Maybe the saturated saline solution reaches a closer value; however, some other materials used in the preparation of mimic phantoms could be used in order to have a better approximation to the values reported in the literature.
- Results sections: the authors report that the measured electrical conductivity is much lower than the physiological value; however, in Introduction, the authors said: we investigated the electrical and thermal conductivities of Layfomm-40 to ensure compatibility with RFA and EAMSs; investigated the use of thermochromic pigments for RFA-sensitivity; and investigated techniques for optimum visibility of Layfomm-40 under X-ray fluoroscopy. How could this huge difference in electrical conductivity affect one of the main goals of this work?
- In Figure 3, the authors said: Both lesions were approximately 3mm in diameter. However, visually, it doesn’t seem both lesions have similar dimensions. Maybe including a rule could help for a better visualization.
- Table 1 must be clearly explained, it is not clear to me how with 80 W the ablation zones can be smaller than the ones generated with lower power. Please, also include time.
- Figure 8 must be clearly explained.
- In conclusion, the authors said: Anatomically accurate, 3D-printed tissue-mimicking thermochromic models, as presented in this paper, may prove to be a reliable, inexpensive and clinically useful tool in simulating cardiac catheter ablation for either training of healthcare professionals or evaluation of novel RF ablation devices. ¿What do you mean by inexpensive? , ¿inexpensive, compared with what?
Reviewer 2 Report
In the Manuscript with title of Proof of Concept of Cardiac Radiofrequency Ablation Simulation using a 3D-Printed Thermochromic Mod. This Manuscript could be acceptable for publication but it needs revisions to help the authors and ensure the quality of the published papers in this journal.
· The paper is generally need to be proofread, formatted and revised for sentence construction and language errors.
· The title of the Manuscript needs to be revised.
· The conclusions section does not really excite the reader that the work presented in this study is new and interesting – the conclusion should be improved, please compile the conclusion again, emphasizing the key outcomes of this study. The authors concluded that these provide a valuable alternative to computer simulations, animal models or cadavers. This needs more details and explanations.
· Please define all abbreviations such as CARTO3 etc
· There are 14 references in the paper given in total. However, a large part of the references was cited in the introduction. There is no references in discussion section. There is no real discussion. It need to be revised. The results should be compared with previous and published results. Not only description.
· Proposed mechanism Mechanisms of action of thermochromic microcapsules during heating and cooling is not clear. It should be revised and explained in more details and included related references from previous literature. (No single reference from previous literature related to the mechanism was included.
· Please Include error bar for all results and how many times that you repeat your experiment.
·
Round 2
Reviewer 2 Report
The paper has been improved and now can be published after some improvemnt in research design and methdology.
